# Integrating health services for HIV infection, diabetes and hypertension in sub-Saharan Africa: a cohort study

Josephine Birungi,[1,2] Sokoine Kivuyo,[3] Anupam Garrib,[4] Levicatus Mugenyi,[2] Gerald Mutungi,[5] Ivan Namakoola,[1] Janneth Mghamba,[6] Kaushik Ramaiya,[7] Duolao Wang [ORCID],[4] Sarah Maongezi,[6] Joshua Musinguzi,[8] Kenneth Mugisha,[2] Bernard M Etukoit,[2] Ayoub Kakande,[1] Louis Wihelmus Niessen,[9] Joseph Okebe,[9] Tinevimbo Shiri,[9] Shimwela Meshack,[10] Janet Lutale,[11] Geoff Gill,[12] Nelson Sewankambo,[13] Peter G Smith,[14] Moffat J Nyirenda,[1,15,16] Sayoki Godfrey Mfinanga [ORCID],[3,9] Shabbar Jaffar [ORCID] [9]

JB, SK and AG contributed equally.
SGM and SJ contributed equally.

JB, SK and AG are joint first authors.
SGM and SJ are joint senior authors.

For numbered affiliations see end of article.

**Correspondence to**
Professor Shabbar Jaffar;
shabbar.jaffar@lstmed.ac.uk

## ABSTRACT

**Background** HIV, diabetes and hypertension have a high disease burden in sub-Saharan Africa. Healthcare is organised in separate clinics, which may be inefficient. In a cohort study, we evaluated integrated management of these conditions from a single chronic care clinic.

**Objectives** To determined the feasibility and acceptability of integrated management of chronic conditions in terms of retention in care and clinical indicators.

**Design and setting** Prospective cohort study comprising patients attending 10 health facilities offering primary care in Dar es Salaam and Kampala.

**Intervention** Clinics within health facilities were set up to provide integrated care. Patients with either HIV, diabetes or hypertension had the same waiting areas, the same pharmacy, were seen by the same clinical staff, had similar provision of adherence counselling and tracking if they failed to attend appointments.

**Primary outcome measures** Retention in care, plasma viral load.

**Findings** Between 5 August 2018 and 21 May 2019, 2640 patients were screened of whom 2273 (86%) were enrolled into integrated care (832 with HIV infection, 313 with diabetes, 546 with hypertension and 582 with multiple conditions). They were followed up to 30 January 2020. Overall, 1615 (71.1%)/2273 were female and 1689 (74.5%)/2266 had been in care for 6 months or more. The proportions of people retained in care were 686/832 (82.5%, 95% CI: 79.9% to 85.1%) among those with HIV infection, 266/313 (85.0%, 95% CI: 81.1% to 89.0%) among those with diabetes, 430/546 (78.8%, 95% CI: 75.4% to 82.3%) among those with hypertension and 529/582 (90.9%, 95% CI: 88.6 to 93.3) among those with multimorbidity. Among those with HIV infection, the proportion with plasma viral load <100 copies/mL was 423(88.5%)/478.

**Conclusion** Integrated management of chronic diseases is a feasible strategy for the control of HIV, diabetes and hypertension in Africa and needs evaluation in a comparative study.

### Strengths and limitations of this study

► This study is a multisite and is the largest ever conducted on integrating chronic care for HIV, diabetes and management in Africa.
► Study was done in close to real-life health service conditions and so findings are likely to be generalisable.
► The study was intended to determine feasibility and early clinical outcomes and does not have a comparison group.

## BACKGROUND

The prevalence of diabetes and hypertension has risen sharply in Africa, where these conditions are now responsible for around 2 million premature adult deaths annually.[1–6] There are effective treatments for diabetes and hypertension, but only 10%–20% of persons living with these conditions are thought to be in regular care,[1 7 8] largely because service provision is limited.[9–11] Identifying approaches to increase access to health services for the management of diabetes and hypertension for populations in Africa represents one of the greatest public health challenges of our time.

Health services in most African countries are generally designed for dealing with acute infections and experience of chronic care management is limited,[12] except for HIV infection which, like other chronic diseases, requires life-long management. In Eastern and Southern Africa 83% of people who know they have HIV infection are in regular care, with 90% of these virally suppressed,[13] compared with a retention rate of about 50% a decade or so ago.[14 15] HIV-associated mortality has declined from over 2 million a

year to less than half a million a year[13] and hypertension and diabetes will need to achieve similar levels of retention and disease control to achieve the large declines in mortality needed to meet the Sustainable Development Goals.[16]

One reason why HIV services have been successful might be because they have been organised in standalone clinics in most African countries from shortly after the time that effective antiretroviral drugs became available. People living with HIV infection are managed in separate primary care clinics, with separate drug and diagnostic procurement channels and a separate funding stream. Now that the burden of other chronic conditions is a rapidly increasing public health problem, how should African health systems organise their services?

Managing multiple chronic conditions in the same clinics could reduce the duplication of healthcare resources and, crucially, could enable the learning acquired by HIV treatment programmes to be applied to the control of other chronic conditions. However, there is little evidence to support this in public health facilities.[17 18] Integrated management could deter people with diabetes and hypertension in seeking care because of the stigma associated with HIV infection. There is also a danger, that expanding the focus of healthcare provision in relatively weak health systems, could put at risk the gains achieved by HIV control programmes. In Tanzania and Uganda, working in close collaboration with policymakers and senior disease control managers, we conducted a cohort study of the integrated care for HIV infection, diabetes and hypertension and evaluated its feasibility and acceptability in terms of retention in care and clinical indicators including control of blood pressure and fasting plasma glucose.

## METHODS
### Study design and setting
We conducted a prospective cohort study, known as the MOCCA (Management of Chronic Conditions in Africa) study. We provided integrated care at 10 health facilities that were offering primary healthcare services, 5 in the Dar es Salaam region in Tanzania and 5 in and around Kampala, Uganda and followed up a cohort of participants over time to determine the feasibility and acceptability of this approach in terms of retention in care and clinical indicators. Online supplemental appendix 1 describes the health facilities included.

The facilities were serving largely urban and periurban populations and chosen purposely. Eight were government run and two were run by non-government organisations. They ranged from regional hospitals through to smaller health centres and dispensaries.[19 20] The larger facilities operated stand-alone separate clinics for diabetes and hypertension, while in the small facilities, patients with diabetes or hypertension were usually seen in the outpatient clinic. Each facility had a HIV clinic run separately from the rest of the clinical services. Patients attending HIV services had their own reception and waiting area, were seen by dedicated clinical staff and had a separate pharmacy. Medical records, track and trace procedures and appointments also all differed from systems employed in other clinics in the facility.

### What was integrated care?
Integration involved the formation of a single clinic for the management of patients with HIV infection, diabetes or hypertension or combinations of these conditions. All patients were seen by the same clinical staff. They shared a single reception, waiting areas and pharmacy. For patients with diabetes or hypertension, systems for tracking and follow-up of patients, recording of clinical notes, counselling, appointments and registrations were mostly set up from new and aligned with those available for people living with HIV, such that all patients were managed in a similar way regardless of their condition.

### How was integrated care implemented?
In Uganda, the first person received integrated care on 5 October 2018 and recruitment finished on 5 April 2019. The final follow-up was on 20 December 2019. In Tanzania, the first person received integrated care on 20 February 2019. Recruitment finished on 23rd June 2019. The final follow-up was on 30 January 2020.

We conducted assessments of each facility using a modified WHO Service Availability and Readiness Assessment tool and reviewed routine data from the Health Management Information Systems (District Health Information System 2). This provided us with knowledge on the clinical infrastructure, availability of medicines, frequency and nature of patient attendance for assessments.

The integrated care clinic was set up to run separately as a stand-alone clinic at each facility so that patients who did not want integrated care would be able to continue with usual clinics. Organisation of the integrated care clinic, including the frequency of clinic days, was done by health facility managers. Where possible the integrated clinic was run on a different day from the HIV and hypertension or diabetes stand-alone clinics. In those facilities where this was not possible, it was run in a separate room away from the main stand-alone clinics. The integrated clinic was held initially on 1 day a week at all facilities and later increased to 2 days a week as the numbers of recruited participants increased. Clinical management was done by healthcare staff employed by the facilities and in accordance with local clinical practice and national guidelines.

The healthcare staff received refresher training in all three chronic health conditions to ensure a common level of understanding of clinical management. Certified national trainers conducted training in both countries. The training comprised a combination of 2 days of classroom sessions and on-the-job training. The doctors specialised in HIV and in non-communicable disease management conducted joint clinics for up to 1 month until they felt comfortable to manage all three

conditions. Classroom training included role play of different scenarios.

Counselling was generally not available for diabetes and hypertension management in either country prior to the study but was available routinely for patients with HIV infection. We therefore provided training to nurses and counsellors on diabetes and hypertension, focused on the need for high treatment adherence to prevent complications. It comprised 1 day of classroom training followed by on-the-job training for up to 4 weeks, as done for clinicians, and mentorship.

Prior to the study, HIV clinics had a system of tracking patients who did not attend a scheduled appointment. This generally involved contacting patients by phone if they missed an appointment, to determine the reason for the missed appointment and to reschedule. Such practices were not in place for people with either diabetes or hypertension. Therefore, we extended the system of tracking patients to cover all three conditions.

All health facility staff received training on the protocol and research methods. This was approximately 1 day of classroom training, followed by on-the-job training.

## Data collection

At each clinic, the research team employed one data clerk and one research nurse to inform patients about the research, collect research data and take informed consent. In addition, small amounts of salary (salary top-ups) were provided to selected facility staff working with study participants and to the head nurse responsible for the facility operations including overseeing the logistical set up of integrated care activities. Research staff worked alongside the health facility staff.

## Selection of participants

In each disease-specific clinic a health talk was given to introduce the study. The number of patients with diabetes or with multiple conditions was very few. Therefore, all persons presenting with diabetes or with two or all three of our target conditions were invited to join the study. Both existing and newly diagnosed patients were enrolled. In the larger facilities (n=3, all in Tanzania), the number of people in care with HIV infection or with hypertension was large and so we sampled systematically (usually every 20th patient) until the clinic ended for the day. In the remaining clinics, where patient load was lower and it was difficult to ascertain the order that patients had come to the facility, we enrolled patients who approached study staff to find out more, or clinicians approached patients and invited them to be screened for the study. On most clinic days, patients who had recently arrived at the facility came forward for enrolment, and those who had nearly completed their consultations for the day and were close to leaving the facility for home, were reluctant to come forward.

## Provision of medicines for hypertension and diabetes

Before the study started, we recognised that supply of medicines for diabetes and hypertension was erratic in both countries. In Uganda, medicines for these conditions were normally provided free but shortages were common. In Tanzania, most patients were required to purchase medicines, and some had health insurance. We have discussed elsewhere the ethical challenges of conducting research of this kind in situations of limited medicines supply.[21]

To ensure a reliable supply of drugs, we purchased back-up supplies of basic drugs for diabetes and hypertension in both Tanzania and Uganda and supplemented patient supplies when the health service stock ran out. The medicines supplied were predominantly first-line drugs and differed between countries depending on what was needed.

Due to the limited supply of medicines for diabetes and hypertension in Uganda, patients at one of the study facilities had organised themselves into a group that met and pooled funds on a monthly basis. These funds were used to purchase the medicines when the health facility ran out of stock. At this facility, we offered minimal buffer supplies compared with other facilities.

## Measurements

Retention was assessed through routine attendance at clinics. Patients who failed to attend an appointment were phoned to establish reasons for not attending and to reschedule. Three attempts were made to contact the patient by phone in the month following non-attendance at an appointment. A patient was regarded as lost to follow-up if they had not attended an appointment within the last 6 months.

Blood pressure was measured using the Omron M6 comfort (OMRON Healthcare, Japan) in Uganda and in three of the facilities in Tanzania; Suresign SLD3-107 (CIGA Healthcare Ltd, UK) was used in the remaining facilities. We measured blood pressure at three time points in the sitting position, 5 min apart, and took the average of the three readings as the pressure. Hypertension was defined as a systolic pressure ≥140 mm Hg and/or a diastolic pressure ≥90 mm Hg and/or use of antihypertensive medication.

Plasma viral load testing was done by the Central Public Health laboratories in Tanzania and Uganda using the Cobas 8800 system (Roche). In Uganda, if the viral load result from the central laboratory was not available or was more than 6 months old, a test was done at the MRC/UVRI/LSHTM Uganda laboratory using Cobas Ampliprep. In both countries glycated haemoglobin (HbA1c) testing was not routinely available in the public health system, and this was done by the study. In Uganda, HbA1c was tested by Cobas 6000 (Roche, Switzerland) at the MRC/UVRI/LSHTM Uganda laboratory, and in Tanzania at the Shree Hindu Mandal Hospital laboratory using a Siemens DCA Vantage (Siemens Healthcare Diagnostics, Germany). Fasting blood glucose was done using

a point of care test in Tanzania (GlucoDr. Auto AGM-4000 and all Medicus Co, Republic of Korea) and in Uganda Contour Plus (Bayer, Germany).

In Tanzania, weight and height were measured using SECA RGZ-160 (SECA, GmbH & Co, Germany). In Uganda, we used an ADE spring scale (1–150 kg) for weight measurement and SECA 213 portable stadiometer (SECA, German) for height measurement. Weight and height were measured to the nearest ±0.1 kg and ±0.1 cm, respectively, and body mass index was calculated as weight/height$^2$ (kg/m$^2$).

### Patient and public involvement

The concept of integrated care was developed in partnership with senior policymakers. In each country, we held planning meetings with the Ministry of Health and nongovernmental partners at national and local level, with senior clinicians and hospital managers and with patient leaders and community representatives. National steering committees were formed, comprising representatives from these stakeholders together with the researchers and these committees contributed to the research strategy and guided its implementation. We also formed an international steering committee, that included four senior researchers and four independent researchers, which had oversight of the whole programme. We also held investigator meetings involving all of the partners and these included patient representatives and policymakers.

### Statistical analysis

The measure of retention used was defined as number in care at study end, calculated as the number enrolled into integrated care minus any deaths, losses to-follow-up, transfers away from the health facility and withdrawals of consent.

Categorical variables were summarised using frequencies and proportions and compared using $\chi^2$ tests. Continuous variables were summarised using means or medians and compared by a t-test or a Wilcoxon test, as appropriate. Generalised estimating equations were used to account for clustering at the health facility level. Kaplan-Meier curves and Cox regression with frailties (to adjust for clustering) were used to analyse retention times. The time was calculated as number of months from first attendance at the integrated clinic to the earliest of the end of the study, loss to follow-up, death, withdrawal or transfer out. The analysis was done using STATA V.15.

### RESULTS

Figure 1 shows the consort chart. Of 2640 patients assessed for eligibility, 102 (3.9%) did not meet eligibility criteria (34 were sick and requiring urgent medical attention or hospital admission, 50 were planning to move out of the area, 13 were pregnant and receiving care from antenatal care services, 5 other reasons), 19 (0.7%) declined and gave no reason, 51 (1.9%) wanted more time to decide (mostly to discuss with their family), 34 (1.3%) were approached just as they had finished consultation and were leaving the clinic and 18 (0.7%) gave other reasons (mostly that they needed to be in specialist care).

Thus, of all those initially assessed for eligibility 91.5% (2416/2640) were invited to join integrated care. This invitation was given at their current clinics and they were asked to return at their next visit to the integrated care

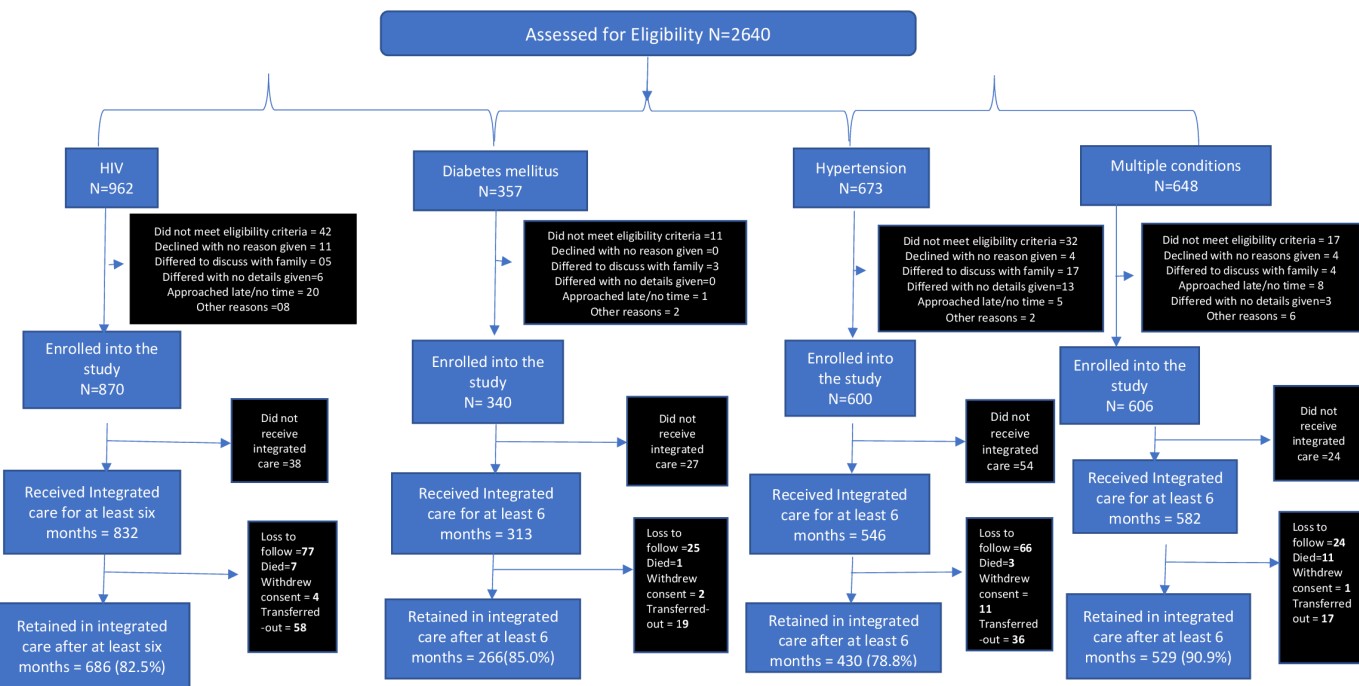

**Figure 1**  Study schema.

**Table 1** Baseline characteristics of patients studied

| Number recruited into integrated care | HIV only | Diabetes only | Hypertension only | Multiple conditions |
|---|---|---|---|---|
| Total | 832 | 313 | 546 | 582 |
| In Tanzania | 352 | 164 | 160 | 251 |
| In Uganda | 480 | 149 | 386 | 331 |
| Type of health facility | | | | |
| Large public facilities | 235 | 81 | 153 | 184 |
| Small public facilities | 409 | 155 | 282 | 217 |
| NGO facilities | 188 | 77 | 111 | 181 |
| Time in care prior to enrolment, number (%)* | | | | |
| <6 months | 310 (37.4) | 63 (20.3) | 165 (30.3) | 39 (6.7) |
| 6–11 months | 41 (5.0) | 28 (9.0) | 42 (7.7) | 22 (3.8) |
| 12–59 months | 167 (20.1) | 98 (31.6) | 172 (31.6) | 135 (23.2) |
| ≥60 months | 311 (37.5) | 121 (39.0) | 166 (30.5) | 386 (66.3) |
| Median (range) months in care | 26.1 (0.0–343.4) | 38.3 (0.0–431.6) | 26.4 (0.0–566.2) | 106 (0.0–534.8) |
| Female, number (%) | 571 (68.6) | 208 (66.5) | 399 (73.1) | 437 (75.1) |
| Median age, (range) | 38 (18–78) | 49 (19–77) | 56 (27–86) | 55 (21–85) |
| BMI, number (%) | | | | |
| <25 | 531 (63.8) | 118 (37.7) | 175 (32.1) | 163 (28.0) |
| 25–29.9 | 185 (22.2) | 98 (31.3) | 181 (33.2) | 202 (34.7) |
| ≥30 | 116 (13.9) | 97 (31.0) | 190 (34.8) | 217 (37.3) |
| Median BMI (range) in men | 21.1 (14.8–35.6) | 24.3 (15.1–37.0) | 25.1 (16.7–41.6) | 26.0 (17.0–53.1) |
| Median BMI (range) in women | 24.4 (13.2–48.9) | 28.5 (15.4–48.2) | 28.8 (16.0–65.2) | 28.8 (15.8–57.9) |

*Missing for seven participants (three with HIV infection, three diabetes and one hypertension).
BMI, body mass index; NGO, non-governmental organisation.

clinic. Six per cent (143/2416) did not return and thus 2273 (86.1%)/2640 were enrolled into integrated care, 927 in Tanzania and 1346 in Uganda.

Table 1 shows the characteristics of the participants at the start of integrated care, for each clinical condition. Overall, 74.5% (1689/2266) reported that they had been in care for 6 months or more prior to the start of the study. For people with multiple conditions, previous time in care was substantially greater than for those with single conditions. Overall, 71.1% (1615/2273) of those included in the study were female.

Median age varied substantially between the different conditions, with that for those living with HIV infection being more than 10 years younger than participants in the other groups (p=0.0001, Kruskal-Wallis test). Obesity was more common among participants with diabetes, hypertension or multimorbidity than among participants with HIV (504/1441 (35.0%) vs 116/832 (13.9%), risk ratio adjusted for clustering: 2.50, 95% CI: 1.88 to 3.31, p<0.001). It was also substantially higher in women than in men (554/1615 (34.3%) vs 66/658 (10.0%), adjusted risk ratio: 3.40, 95% CI: 2.63 to 4.41, p<0.001).

Table 2 shows the biomedical indicators as measured at baseline on the day of integration or within the previous 6 months. The majority did not reach target blood pressure and glycaemia control levels, whether they had a single condition or multimorbidity. When participants who reported that they had been in care for less than 6 months (ie, patients who had newly started therapy) were excluded, 509 (57.7%)/882 of those with hypertension had a blood pressure ≥140/90 mm Hg, 355/508 (69.9%) of those with diabetes had a fasting blood glucose ≥7.0 mmol/L and 72 (16.4%)/438 of those with HIV infection had a plasma viral load ≥100 copies/mL.

The proportions alive and retained in care were high among all participants (table 3). The overall proportion alive and in care among participants with a single condition was 1382/1691 (81.7%, 95% CI: 79.9% to 83.6%), which did not differ significantly from the proportion with multimorbidity (adjusted risk ratio: 0.91, 95% CI: 0.82 to 1.00, p=0.061).

Figure 2 shows the survival probability curves of being alive and in care over time. After adjusting for clustering, the probability (95% CI) of being retained and alive in care at 12 months was 0.74 (0.66–0.80) among participants with HIV alone, 0.74 (0.66–0.81) among those with diabetes alone, 0.70 (0.64–0.76) among those with hypertension alone and 0.84 (0.79–0.88) among those with multimorbidity (p=0.008).

Table 4 shows the biomedical indicators of participants at the final follow-up. The proportions of patients living with HIV infection with viral suppression were

**Table 2** Biomedical indicators at baseline

| | Patients who have a single condition occurring alone | The condition occurs with other conditions, causing multimorbidity (only among those with the condition) | Both combined |
|---|---|---|---|
| **Among patients with hypertension** | | | |
| Blood pressure (mm Hg), number (%) | n=546 | n=539 | n=1085 |
| ≥140/90 | 342 (62.6) | 327 (60.7) | 669 (61.7) |
| <140/90 | 204 (37.4) | 212 (39.3) | 416 (38.3) |
| ≥180/120 | 42 (7.7) | 64 (11.9) | 106 (9.8) |
| <180/120 | 504 (92.3) | 475 (88.1) | 979 (90.2) |
| Systolic BP mm Hg, mean (range) | 144.4 (83.0–257.7) | 145.1 (68.7–271.0) | 144.7 (68.7–271.0) |
| Diastolic BP mm Hg, mean (range) | 90.2 (43.3–163.0) | 90.3 (54.7–167.3) | 90.2 (43.3–167.3) |
| **Among patients with diabetes** | | | |
| Fasting glucose (mmol/L), number (%) | n=228 | n=281 | n=509 |
| <6.1 | 34 (14.9) | 62 (22.1) | 96 (18.9) |
| 6.1–6.9 | 17 (7.5) | 32 (11.4) | 49 (9.6) |
| ≥7.0 | 177 (77.6) | 187 (66.6) | 364 (71.5) |
| Fasting glucose, mean (range) mmol/L | 11.5 (2.9–29.3) | 9.8 (2.3–28.2) | 10.5 (2.3–29.3) |
| HbA1c, number (%)* | n=124 | n=206 | n=330 |
| <6.0% | 4 (3.2) | 25 (12.1) | 29 (8.8) |
| 6.0%–6.4% | 7 (5.7) | 22 (10.7) | 29 (8.8) |
| ≥6.5% | 113 (91.1) | 159 (77.2) | 272 (82.4) |
| HbA1c, mean (range) in % | 10.1 (4.8–16.9) | 8.8 (4.8–16.5) | 9.3 (4.8–16.9) |
| **Among patients living with HIV** | | | |
| HIV plasma viral load, number (%)* | n=493 | n=153 | n=646 |
| <100 copies/mL | 306 (62.1) | 131 (85.6) | 437 (67.7) |
| 100–999 copies/mL | 68 (13.8) | 14 (9.2) | 82 (12.7) |
| ≥1000 copies/mL | 119 (24.1) | 8 (5.2) | 127 (19.7) |

Note: blood pressure was measured at baseline by research staff among 1187 (99.9%)/1188 participants who were not known previously to be hypertensive. Of these, the proportion who were detected with a blood pressure of 140/90 mm Hg or higher was 80 (9.6%)/831 among those with HIV infection alone, 47 (15.0%)/313 among those with diabetes alone and 6 (14.0%)/43 among those with HIV infection and diabetes. Of note, among all HIV-infected participants, blood pressure was 180/120 mm Hg or higher among 45 (4.0%)/1126.
*Measured at any time within the previous 6 months. Overall for the proportion with plasma viral load <100 copies/mL, the 95% CI was 64.1%, 71.4% and 80.3% (95% CI: 77.3% to 83.5%) had a plasma viral load <1000 copies/mL.
BP, blood pressure; HbA1c, glycated haemoglobin.

high, but the proportions of those with hypertension achieving blood pressure less than 140/90 mm Hg and those with diabetes achieving glycaemia <7 mmol/L were substantially lower. However, blood pressure, fasting glucose and plasma viral load were all better controlled at study end than at baseline (table 5), even when participants who had been in care for less than 6 months, and would have been newly started on therapy were excluded. After adjusting for clustering, among patients with HIV, the proportion (95% CI) with plasma viral load of <100 copies/mL was 88.5% (85.7%–91.4%) and those with plasma viral load <1000 copies/mL was 96.0% (94.3%–97.8%).

## DISCUSSION

This study shows that integrating care for HIV infection, diabetes and hypertension achieved high levels of retention in care for people living with diabetes or hypertension in Africa. Crucially, this improvement did not appear to be at the expense of HIV control, as retention in all three groups of people with single conditions was similar, and the proportion with an HIV viral suppression of less than a 100 copies/mL was close to 90%.

Thus, our study suggests that an integrated approach could achieve excellent retention for diabetes and hypertension in an integrated clinic, as has been achieved for

**Table 3** Retention in integrated care and clinical outcomes according to condition

| Overall | HIV only | Diabetes only | Hypertension only | Multiple conditions |
|---|---|---|---|---|
| Number enrolled into integration | 832 | 313 | 546 | 582 |
| Median follow-up (interquartile range) | 8.2 (6.9–10.4) | 8.0 (6.9–9.4) | 8.2 (6.9–10.1) | 8.2 (7.2–9.8) |
| Retained in care to the end of the study, n% (95% CI)* | n.686 82.5 (79.9 to 85.1) | n.266 85.0 (81.1 to 89.0) | n.430 78.8 (75.4 to 82.3) | n.529 90.9 (88.6 to 93.3) |
| Rate per 100 person years (95% CI)* | 75.0 (70.6 to 78.7) | 77.7 (70.4 to 83.3) | 69.9 (63.5 to 74.6) | 87.1 (83.2 to 90.2) |

Note: 85 persons were newly diagnosed with multiple conditions over the course of follow-up. Of participants with HIV alone, 5 were diagnosed with diabetes and 33 with hypertension. Of those with diabetes, 3 tested positive for HIV and 23 were diagnosed with hypertension. Of those with hypertension, 1 tested positive for HIV and 9 were diagnosed with diabetes. Eleven persons with dual conditions were diagnosed with a third condition.
*Adjusted for clustering at health facility level.

people living with HIV infection where retention exceeds 80% today, up from about 50% a decade or so ago.[13–15]

The retention rate for people with diabetes or hypertension observed in our study was higher than that published from some other studies, which involved studies of single diseases, including those where diagnostic and treatment services were enhanced and, for example, treatments were provided for free.[22–24] In our study, we put in place basic measures to support participants with diabetes and hypertension including minimal access to basic medicines, counselling and telephone reminders for those who missed appointments. However, these measures were no more than those available to people in care with HIV infection and no more than those available to people in research studies of hypertension and diabetes. No financial or other incentives were provided in our study and participants were managed by routine healthcare staff. Analysis of the costs of this package will be done separately but generic medicines for diabetes and hypertension and associated monitoring costs are substantially less than those for HIV infection and so we anticipate that the model will be highly cost-effective.

In a large study in Malawi conducted by members of our team almost a decade ago, people identified with diabetes and hypertension in the community were referred into care with a dedicated medical officer and access to a free and reliable supply of medicines and diagnostics. However, at 12 months retention in care was 27%, suggesting that control of diabetes and hypertension when these conditions are identified through routine screening (ie, are asymptomatic) might be a challenge.[23 25] A study from Kwazulu-Natal in South Africa reported similar findings.[26] However, other studies that provided a good level of care and that have involved patients who know they have diabetes and hypertension have also struggled to achieve good levels of retention.[22]

Why did our study achieve high retention? For people with multiple conditions, a huge advantage of integrated care is that they only need to attend one clinic appointment, but in our study retention was also high for people with single conditions. It is possible that when services were integrated and the lessons learnt in managing HIV infection with respect to counselling, treatment adherence and the provision of a more streamlined service, were applied by healthcare staff to diabetes and hypertension management, patients responded to the changes in service delivery. The mixing of patients with HIV infection with those living with diabetes and hypertension may also have led to the sharing of experiences of living with a long-term condition, which could have brought further benefit.

We knew that integration of services would involve expansion of care responsibilities of health facility staff, and feared that this would adversely impact HIV care outcomes.[17 27 28] However, this did not happen. HIV viral suppression rates were high at enrolment and were slightly higher at the end of the study. There is an increasing body of literature on integrating screening, diagnosis and treatment for non-communicable diseases into established HIV services.[17 29 30] As far as we are aware, ours is the first study to test integration of services fully so that any person with one or more of the three target conditions could be managed in one clinic.[29 31]

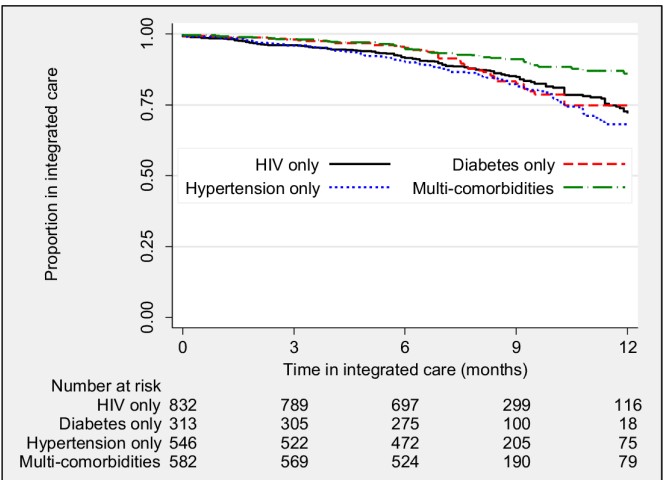

**Figure 2** Kaplan-Meier curves showing probability proportion of patients of remaining in care according to condition.

| Table 4 Biomedical indicators at study end | | | |
|---|---|---|---|
| | Patients who have a single condition occurring alone | The condition occurs with other conditions, causing multimorbidity | Both combined |
| Blood pressure (mm Hg), number (%) | n=415 | n=469 | n=884 |
| ≥140/90 | 183 (44.1) | 225 (48.0) | 408 (46.2) |
| <140/90 | 232 (55.9) | 244 (51.2) | 476 (53.9) |
| ≥180/120 | 7 (1.7) | 25 (5.3) | 32 (3.6) |
| <180/120 | 408 (98.3) | 444 (94.7) | 852 (96.4) |
| Systolic BP mm Hg, mean (range) | 133.9 (26.0–228) | 136.7 (90.0–227) | 135.4 (26–228) |
| Diastolic BP mm Hg, mean (range) | 83.9 (53–130) | 85.6 (44–185) | 84.8 (44–185) |
| Fasting glucose (mmol/L), number (%) | n=189 | n=242 | n=431 |
| <6.1 | 36 (19.1) | 69 (28.5) | 105 (24.4) |
| 6.1–6.9 | 28 (14.8) | 33 (13.6) | 61 (14.2) |
| ≥7.0 | 125 (66.1) | 140 (57.9) | 265 (61.5) |
| Fasting glucose, mean (range) mmol/L | 9.7 (3.6–28.5) | 8.3 (3.6–19.8) | 8.9 (3.6–28.5) |
| HbA1c, number (%). | n=84 | n=140 | n=224 |
| <6% | 17 (20.2) | 31 (22.1) | 48 (21.4) |
| 6.0%–6.4% | 4 (4.8) | 15 (10.7) | 19 (8.5) |
| ≥6.5% | 63 (75.0) | 94 (67.1) | 157 (70.1) |
| HbA1c, mean (range) percent | 7.6 (0.6–13.5) | 7.3 (0.6–15.9) | 7.4 (0.6–15.9) |
| HIV plasma viral load, number (%)* | n=322 | n=156 | n=478 |
| <100 copies/mL | 278 (86.3) | 145 (93.0) | 423 (88.5) |
| 100–999 copies/mL | 27 (8.4) | 9 (5.8) | 36 (7.5) |
| ≥1000 copies/mL | 17 (5.3) | 2 (1.3) | 19 (4.0) |

Note: 80 persons were newly diagnosed with second or third conditions over the course of the follow-up (see table 3). Their final follow-up indicators are not included in the analyses above as these patients may have been newly started on treatment and therefore will not have had time to stabilise.
*The 95% CI for plasma viral load <100 copies/mL for both groups combined was 85.7, 91.4%. The proportion with plasma viral load <1000 copies/mL was 96.0% (95% CI: 94.3% to 97.8%).
BP, blood pressure; HbA1c, glycated haemoglobin.

We had feared that stigma may put off patients with diabetes and hypertension from attending a joint service with patients living with HIV. However, none of the patients who declined to join cited stigma. It is possible that the 41 patients who declined without giving a reason or who deferred their decision, did so because of the stigma. Nonetheless, these numbers of refusals are relatively small, and very few study participants withdrew over

| Table 5 Risk differences (95% CI) between baseline and study end among participants living with hypertension, diabetes or HIV infection | | |
|---|---|---|
| Clinical condition (either alone or alongside other conditions) | All participants | Excluding participants in care for less 6 months at enrolment |
| Raised blood pressure (≥140/90 mm Hg) | 14.8%, (10.5% to 19.2%) (p<0.001) | 10.5%, (5.7% to 15.3%) p<0.001 |
| Raised fasting glucose (≥7 mmol/L) | 8.4%, (2.4% to 14.3%) p=0.006 | 7.4% (1.0% to 13.7%) p=0.024 |
| Lack of viral suppression (plasma viral load ≥100 copies/mL) | 20.8%, (16.0% to 25.5%) p<0.001). | 7.3% (2.7% to 11.9%) p=0.002 |

Notes: the risk difference is calculated as risk estimate at baseline minus risk estimate at study end.

time. HIV has a special status in almost all African countries, which may have contributed to the stigma associated with this disease. Our study suggests that HIV infection can be managed in the same space as other conditions—that is, that HIV can be considered like any other disease or condition—and this will not make a difference to the outcome for the patient living with HIV infection.

Although we achieved high rates of retention, and blood pressure and glycaemia control improved over time, a high percentage of those with diabetes or hypertension continued to have uncontrolled blood pressure and uncontrolled glycaemia at the end of the study, in contrast to the situation with HIV for which the vast majority maintained excellent control of viraemia. In the case of diabetes, this is complicated by a lack of access to adequate monitoring. Access to HbA1c testing was a challenge we faced in this study, and is a challenge across Africa.[32] The next step in this story is to identify and test interventions that improve blood pressure and diabetes control (among people in care) and take this control to levels achieved by HIV programmes in order to drive down mortality from non-communicable diseases.

Large numbers of people in high-income countries are now living with multiple conditions—so called multimorbidity—and this number is expected to continue to rise sharply in Africa, if it has not done so. In our study, individuals with more than one condition, who would normally attend multiple different clinics, appeared to have a higher retention in care than people with a single condition but their clinical indicators at study end were no different from those with a single condition. While it is not surprising that their retention should be high, the lack of superiority in their clinical indicators from those with a single condition suggest that greater efforts will be needed to improve clinical outcomes.

Studies that involve reorganisation of care provision represent huge change and challenges for health services, communities and patients. They are not possible without partnerships based on an ethos of equality and openness and without the building of trust between researchers and these stakeholders. It is essential that research of this nature allows time and resource for the working together of different groups of researchers, healthcare providers and public health authorities, patients and communities.

## Strengths and limitations

Although the evidence generated by our study is much greater than that available to most policymakers, and separate clinics for each chronic condition are not sustainable for Ministries of Health, changes to health policy that would affect tens of millions of people living with chronic conditions requires robust evidence. Our study was large, comprising over 2000 participants with different chronic conditions. However, it was a pilot study and did not have a comparison group. It was done in just 10 health facilities, 5 in Tanzania and 5 in Uganda. In some health facilities, the number of people presenting with HIV or hypertension was high and the facilities had no system for recording the order in which patients presented and so we were unable to choose participants systematically but selected them purposively. This could have introduced selection bias.

Evidence is needed from large randomised trials with control groups comprising current standard care as provided in stand-alone clinics,[17] with replication in multiple settings and over a longer duration of follow-up and with systematic or random sampling of participants to reduce the possibility bias. We also need more evidence on why integrated care works for different groups of people either with single or with multiple conditions.

**Author affiliations**
[1]MRC/UVRI and LSHTM Uganda Research Unit, Entebbe, Uganda
[2]The AIDS Support Organization, Kampala, Uganda
[3]Muhimbili Medical Research Centre, National Institute for Medical Research Muhimbili Research Centre, Dar Es Salaam, Tanzania
[4]Department of Clinical Sciences, Liverpool School of Tropical Medicine, Liverpool, UK
[5]Non-Communicable Diseases Control Programme, Ministry of Health, Kampala, Uganda
[6]Ministry of Health, Community Development, Gender, Elderly and Children, Dodoma, Tanzania
[7]Shree Hindu Mandal Hospital, Dar es Salaam, Tanzania
[8]AIDS Control Programme, Ministry of Health, Kampala, Uganda
[9]Department of International Public Health, Liverpool School of Tropical Medicine, Liverpool, UK
[10]Amana Regional Referral Hospital, Dar es Salaam, Tanzania
[11]Muhimbili University of Health and Allied Sciences, Dar es Salaam, Tanzania
[12]Liverpool School of Tropical Medicine, Liverpool, UK
[13]Makerere University College of Health Sciences, Kampala, Uganda
[14]MRC International Statistics and Epidemiology Group, London School of Hygiene & Tropical Medicine, London, UK
[15]Malawi Epidemiology and Intervention Research Unit (MEIRU), Lilongwe and Karonga, Malawi
[16]Faculty of Epidemiology and Public Health, London School of Hygiene and Tropical Medicine, London, UK

**Acknowledgements** We wish to thank Hazel Snell, Erik van Widenfelt, Luis Cuevas, the MOCCA study teams in Tanzania and Uganda and the health facility teams who supported the study and provided patient care, and Drs Mina Nakawuka Ssali and Martha Nabbada from the Ministry of Health in Uganda for their time and support, and Ayubu Masasi, Eric Mgina, Micheal Mubiru and Sebastian Owilla Kidega on data management.

**Contributors** SJ, MJN, ShimwelaM, AG, GM, JannethM, JoshuaM, SarahM, JL, GG, NS and PS designed the study with support from KR, KM, BE and LWN. JB, SK and IN implemented the study with support from AG, ShimwelaM, MJN, JO, TS, and KR. SK and AK managed the data with support from JB and AG. JB, SK and AG wrote the first drafts and SJ led the later drafts. LM did the statistical analysis with support from DW. JB coordinated the study in Uganda and SK in Tanzania, with support from SJ, AG, MJN and ShimwelaM. All authors contributed to various drafts of the paper and saw and approved the final version. The corresponding author is the overall guarantor - prof Shabbar Jaffar.

**Funding** This research was funded by the National Institute for Health Research (NIHR) (project reference 16/137/87) using UK aid from the UK Government to support global health research. The views expressed in this publication are those of the author(s) and not necessarily those of the NIHR or the UK Department of Health and Social Care.

**Competing interests** None declared.

**Patient consent for publication** Consent obtained directly from patient(s).

**Ethics approval** The study received ethical clearance from TASO Research Ethics Committee and Uganda National Council for Science and Technology (UNCST) in Uganda (Reference TASO REC/015/18-UG-REC-009), National Health Research Ethics Review Sub-Committee in Tanzania (Reference NIMR/HQ/R.8a/

Vol. IX/2793), and the Liverpool School of Tropical Medicine in the UK (Reference 18-044).

**Provenance and peer review** Not commissioned; externally peer reviewed.

**Data availability statement** No data are available.

**ORCID iDs**
Duolao Wang http://orcid.org/0000-0003-2788-2464
Sayoki Godfrey Mfinanga http://orcid.org/0000-0001-9067-2684
Shabbar Jaffar http://orcid.org/0000-0002-9615-1588

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
