## [Reviewer comments · BMJ Open]

ARTICLE DETAILS

TITLE (PROVISIONAL)	Integrating health services for HIV-infection, diabetes and hypertension in sub Saharan Africa: a cohort study.
AUTHORS	Birungi, Josephine; Kivuyo, Sokoine; Garrib, Anupam; Mugenyi, Levi; Mutungi, Gerald; Namakoola, Ivan; Mghamba, Janneth; Ramaiya, Kaushik; Wang, Duolao; Maongezi, Sarah; Musinguzi, Joshua; Mugisha, Kenneth; Etukoit, Bernard; Kakande, Ayoub; Niessen, Louis; Okebe, Joseph; Shiri, Tinevimbo; Meshack, Shimwela; Lutale, Janet; Gill, Geoff; Sewankambo, Nelson; Smith, Peter; Nyirenda, Moffat; Mfinanga, Sayoki; Jaffar, Shabbar

VERSION 1 – REVIEW

REVIEWER	Philip Smith University of Cape Town, Desmond Tutu HIV Centre
REVIEW RETURNED	28-Jun-2021

GENERAL COMMENTS	Review The manuscript presents important findings about integrated care for chronic conditions in two countries. Integrated chronic care services may improve care, quality of life, quality of experience of patients, and improve disease outcomes. These findings show that there is evidence to support further investigations, including cost evaluations, of integrating these services. Abstract: - The abstract makes a strong statement, "Health care... is inefficient and wastes resources". This statement needs to be supported by the cited literature, or removed.- The objective of the study needs to be clarified. Introduciton: - Paragraph three requires citations of the introduction requires citations.- Paragraph three states that HIV clinics are successful because they serve one need. Does integrating care move towards a primary health model which may be less effective than clinics that only provide HIV care? This needs to be explored. Methods: - The methods section requires clarity. The objective/ research question should be made clear. Paragraph two in the study design and setting section states that the clinics were chosen purposefully, based on their infrastructure; state what that infrastructure was.- In the "how was integrated care implemented" section: Paragraph 3 – the reason for two models of implementation needs to be clarified. Why was an observation period needed in one set of clinics, and what was considered "sufficient understanding" to implement
---

	integrated care without this observation?  - Paragraph 3 – Line 1 seems to suggest that participants self-selected into the study. This may be a strength or a limitation of the study. Could this introduce bias into the results? - Paragraph 4 – it may be clearer to state to state “...three chronic health conditions...” - Paragraph 4 – sentence could be split up to improve clarity. - Paragraph 4 – it may be beneficial to state this in active language to say, “Certified national trainers conducted training in Uganda, while senior physicians from a non-governmental hospital ran training in Tanzania”, and then describe the training. - It is implied that the national trainers were trained, but were the physicians trained as well? - Selection of participants: this section needs clarity. What was the protocol for recruitment? Was recruitment different for each facility? - Statistical analysis: It may need clarification why the not retained group included deaths, transfers, and withdrawals of consent since these seem to be different from loss-to-follow-up. Discussion:  - Because the objective is unclear, the discussion makes reaching conclusions that may not be supported by the findings. In paragraph 1, since this was not a comparative study, it may be worth stating that the levels of retention in this study were compared with previous studies. - The improved follow up in the form of calls to those with diabetes and hypertension requires discussion, since this may have influenced the retention outcome. It may be informative to state what characteristics may have contributed to the high levels of retention in both patients with single chronic diseases and those with multiple chronic diseases. The light-touch follow-up intervention may have significant impact on retention. Other comments:  - It is general convention to use words to write out numbers under ten.
--	--

REVIEWER	Morgan Birabaharan University of California San Diego
REVIEW RETURNED	23-Jul-2021

GENERAL COMMENTS	There has been a surge in the burden of noncommunicable disease in sub-Saharan African. Therefore, of utmost importance will be to establish new health care delivery models and resources as morbidity and mortality related to infection in sub-Saharan African will soon be outpaced by cardiovascular disease, metabolic disorders, and malignancy. In this report, Birungi et al share their findings in broadening the services of a HIV clinic to also care for those with hypertension and diabetes. Their strategy demonstrates exciting results, showing high retention and improvement in blood pressure and diabetes control. Strengths of this study include large sample size and a novel question that is of high importance. However, the manuscript is largely confusing. Authors should be encouraged about the importance of their paper and the need to disseminate their results, but major revisions are needed. Introduction:
---

	The importance of the paper or hypothesis is not appreciated in the introduction. Why are authors looking to expand services of an HIV clinic, rather than establish new primary care clinics? Are authors suggesting building new primary care clinics to service the increasing burden of noncommunicable diseases is not feasible? And it might be more resourceful to repurpose current HIV clinics to include other services? If so, the authors need to explicitly state this. Authors should look to shorten their introduction, the second paragraph (Lines 16-25) seems unnecessary Also, what is the main question of the paper? Is it that, will people seek care at an integrated health clinic that also serves HIV, despite possible stigma? The lack of a clear question causes confusion throughout the rest of the paper Methods The methods are unfortunately excessive. Authors should consider moving most of methods to supplementary material or re-consider what is necessary to include in the manuscript in order for results to be replicated Page 5, Lines 23-Line 31 seems unnecessary Page 5 Lines 46-Page 6 line 58 – should be moved to supplementary or condensed Page 7 Lines 21-41 – this seems unnecessary to have Retention needs an explicit definition. Is it considered retention if a patient has one follow up visit at 6 months? If blood pressure and glycemic control are important measures of success to this project, there should be mention of this in the introduction Authors should not feel the need to mention specific instruments used to measure A1C, record blood pressures, or measures Page 8 lines 3-22 can largely be removed or condensed Results Authors should re-evaluate what results should be included in the manuscript. Would advise to only include results that are pertinent to their original question. For example, providing relative risk ratios for obesity among those with diabetes vs HIV, seems irrelevant. Figure 1 – too large, should be condensed. Would reason there is no need to stratify by disease Table 2 and Table 4 need to be combined to one table for easy interpretation Discussion Study makes comparisons to other studies that assessed retention. Would be important to state if retention is defined differently in other studies
--	--

	Authors should look to condense discussion. A point to be made for that is in reference to the entire manuscript - it is thought to be stigmatizing to describe patients with HIV, as living with “HIV infection.” Would recommend changing verbiage to reflect the more accepting terminology, “People/Persons/Patients with HIV”. “In our study, individuals with more than one condition, who would normally attend multiple different clinics,” 1. This is a statement that should be in the introduction as it gets across the importance of authors’ work
--	---

VERSION 1 – AUTHOR RESPONSE

Reviewer: 1

Dr. Philip Smith, University of Cape Town

Comments to the Author:

The manuscript presents important findings about integrated care for chronic conditions in two countries. Integrated chronic care services may improve care, quality of life, quality of experience of patients, and improve disease outcomes. These findings show that there is evidence to support further investigations, including cost evaluations, of integrating these services.

Abstract:

- **Comment 1:** *The abstract makes a strong statement, “Health care... is inefficient and wastes resources”. This statement needs to be supported by the cited literature, or removed.*

Response: Ours is the first study to evaluate integrated management for these conditions in East Africa. There is no previous reliable evidence that we could cite to support but it is reasonable to assume that separate clinics would require more resources to run than a single clinic. This is based on experience and on discussions with health care managers.

Nonetheless, as the evidence to support this is limited, we have modified this as follows:

Health care for these conditions is organised in separate stand-alone (vertical) clinics, which may be inefficient and may wastes resources.

- **Comment 2:** *The objective of the study needs to be clarified.*

Response: We have done this. On page 3 in the abstract, we say:

Objectives: To determined the feasibility and acceptability of integrated management of chronic conditions in terms of retention in care and clinical indicators.

Similar text is repeated at the end of the main introduction (p4):

In Tanzania and Uganda, working in close collaboration with policy makers and senior disease control managers, we conducted a cohort study of the integrated care for HIV-infection, diabetes and hypertension and evaluated its feasibility and acceptability in terms of retention in care and clinical indicators.

Introduction:

- **Comment 3:** *Paragraph three requires citations of the introduction requires citations.*

Response: Not sure what the referee is asking. We have modified the first sentence from 'why HIV services have been successful is because' to 'why HIV services have been successful **might be** because'

HIV vertical care has never been compared with integrated care – so there are no references to cite. This is why we conducted a feasibility study as evidence was so scarce.

- **Comment 4:** *Paragraph three states that HIV clinics are successful because they serve one need. Does integrating care move towards a primary health model which may be less effective than clinics that only provide HIV care? This needs to be explored.*

Response. Paragraph 3 discusses the benefits of HIV clinics serving one need, as referee says. The potential challenges of integrated care that the referee requests are in paragraph 4. Bottom of Page 4 says:

Integrated management could deter people with diabetes and hypertension in seeking care because of the stigma associated with HIV-infection. There is also a danger, that expanding the focus of health care provision in relatively weak health systems, could put at risk the gains achieved by HIV control programmes.

Methods:

- **Comment 5:** *The methods section requires clarity. The objective/ research question should be made clear.*

Response: As discussed in the points above, we have clarified the objectives in the abstract and at the end of the main introduction on page 4

- **Comment 6:** *Paragraph two in the study design and setting section states that the clinics were chosen purposefully, based on their infrastructure; state what that infrastructure was.*

Response: What we meant was that we chose a range of health facilities. We have corrected this now. It reads:

Eight were government run and 2 were run by non-government organisations. They ranged from regional hospitals through to smaller health centres and dispensaries [19, 20].

- **Comment 7:** *In the "how was integrated care implemented" section: Paragraph 3 – the reason for two models of implementation needs to be clarified. Why was an observation period needed in one set of clinics, and what was considered "sufficient understanding" to implement integrated care without this observation?*

RESPONSE. We have clarified this. On the top of page 5, we now say:

We had planned a brief observation period prior to implementing integrated care so to understand better the procedures at each clinic, such as the patient flow. We did this in the first 6 facilities. Very few patients had declined to join

and thereafter we felt confident on how to set up the integrated care. Therefore, in the remaining 4 facilities, implementation was done straightaway without an observation period.

- **Comment 8:** *Paragraph 3 – Line 1 seems to suggest that participants self-selected into the study. This may be a strength or a limitation of the study. Could this introduce bias into the results?*

Response: Yes, there was some self-selection among patients presenting with HIV or hypertension in some of the facilities. We recognise this as a limitation and have discussed this in the Discussion. On page 12, we say

In some health facilities, the number of people presenting with HIV or hypertension was high and the facilities had no system for recording the order in which patients presented and so we were unable to choose participants systematically but selected them purposively. This could have introduced selection bias

In the final paragraph of the Discussion on page 12. We say evidence is needed from studies that sample systematically or randomly to avoid bias.'

- **Comment 9:** *Paragraph 4 – it may be clearer to state to state "...three chronic health conditions..."*

Response: We have done this. Page 6, para 4 reads training in all three chronic health conditions ...

- **Comment 10:** *Paragraph 4 – sentence could be split up to improve clarity.*

Response: We have done this. This now reads (page 6)

The doctors specialised in HIV and in non-communicable disease management conducted joint clinics for up to one month until they felt comfortable to manage all three conditions.

- **Comment 11:** *Paragraph 4 – it may be beneficial to state this in active language to say, "Certified national trainers conducted training in Uganda, while senior physicians from a non-governmental hospital ran training in Tanzania", and then describe the training.*

Response: We have re-arranged this paragraph as you suggested. It now reads:

The health care staff received refresher training in all three chronic health conditions to ensure a common level of understanding of clinical management. Certified national trainers conducted training in both countries. The training comprised a combination of two days of classroom sessions and on-the-job training. The doctors specialised in HIV and in non-communicable disease management conducted joint clinics for up to one month until they felt comfortable to manage all three conditions. Classroom training included role play of different scenarios.

Comment 12: *It is implied that the national trainers were trained, but were the physicians trained as well?*

Response: We have corrected this – see point 11 response above.

- **Comment 13:** *Selection of participants: this section needs clarity. What was the protocol for recruitment? Was recruitment different for each facility?*

Recruitment had to differ by disease and by size of facility.

The numbers of patients presenting with diabetes or multiple conditions was few and so we enrolled everyone.

The patients presenting with hypertension or HIV were many and so we sampled. In large facilities, this was easy to do as they recorded the order that patients presented. In other facilities, this information was not available.

This is explained under selection of participants (page 7).

- **Comment 14:** *Statistical analysis: It may need clarification why the not retained group included deaths, transfers, and withdrawals of consent since these seem to be different from loss-to-follow-up.*

Response. There are many ways in which retention can be defined. We defined it apriori. This is stated in the methods and have done the analysis with this definition.

Discussion:

- **Comment 15:** *Because the objective is unclear, the discussion makes reaching conclusions that may not be supported by the findings. In paragraph 1, since this was not a comparative study, it may be worth stating that the levels of retention in this study were compared with previous studies.*

Response: The objectives of the study have now been stated both in the abstract and in the background sections. We start the Discussion with the main findings and then immediately compare with other studies (e.g. with HIV – references 13-15).

- **Comment 16:** *The improved follow up in the form of calls to those with diabetes and hypertension requires discussion, since this may have influenced the retention outcome.*

Response: We did not just provide telephone reminders to people with diabetes and hypertension, but these participants had access to basic medicines and received counselling. It is not possible to say to what extent each factor influenced retention.

Please note that patients with diabetes and hypertension had the same telephone reminders and access to medicines and counselling as patients with HIV were already getting. Nothing special was done for people with diabetes and hypertension. The chronic clinic essentially provided the same package to anyone with a chronic condition.

This is clearly stated on page 11:

In our study, we put in place basic measures to support participants with diabetes and hypertension including minimal access to basic medicines, counselling and telephone reminders for those who missed appointments. However, these measures were no more than those available to people in care with HIV-infection and no more than those available to people in research studies of hypertension and diabetes. No financial or other incentives were provided in our study and participants were managed by routine health care staff.

Comment 17: *It may be informative to state what characteristics may have contributed to the high levels of retention in both patients with single chronic diseases and those with multiple chronic diseases. The light-touch follow-up intervention may have significant impact on retention.*

Response: We have speculated and did do social science studies which form separate papers. It is difficult to go beyond our speculation as in complex interventions, there are many factors involved.

This is what we say on page 11.

.... but in our study retention was also high for people with single conditions. It is possible that when services were integrated and the lessons learned in managing HIV-infection with respect to counselling, treatment adherence and the provision of a more streamlined service, were applied by health care staff to diabetes and hypertension management, patients responded to the changes in service delivery. The mixing of patients with HIV-infection with those living with diabetes and hypertension may also have led to the sharing of experiences of living with a long-term condition, which could have brought further benefit.

Other comments:

- **Comment 18:** *It is general convention to use words to write out numbers under ten.*

Response: Thank you. We were not aware of this but happy to change numbers in text to words.

Reviewer: 2

Dr. Morgan Birabaharan, University of California San Diego

Comments to the Author:

There has been a surge in the burden of noncommunicable disease in sub-Saharan African. Therefore, of utmost importance will be to establish new health care delivery models and resources as morbidity and mortality related to infection in sub-Saharan African will soon be outpaced by cardiovascular disease, metabolic disorders, and malignancy. In this report, Birungi et al share their findings in broadening the services of a HIV clinic to also care for those with hypertension and diabetes. Their strategy demonstrates exciting results, showing high retention and improvement in blood pressure and diabetes control.

Strengths of this study include large sample size and a novel question that is of high importance. However, the manuscript is largely confusing. Authors should be encouraged about the importance of their paper and the need to disseminate their results, but major revisions are needed.

Introduction:

The importance of the paper or hypothesis is not appreciated in the introduction. Why are authors looking to expand services of an HIV clinic, rather than establish new primary care clinics? Are authors suggesting building new primary care clinics to service the increasing burden of noncommunicable diseases is not feasible? And it might be more resourceful to repurpose current HIV clinics to include other services? If so, the authors need to explicitly state this.

Response: We have clarified the objectives in both the abstract and the introduction, as discussed above responding to the editor and reviewer 1 comments.

We also state in the methods how the integrated clinics were set up and how they ran as separate entities within the health facilities. This is on page 5. It includes a sentence that says

“The integrated care clinic was set up to run separately as a stand-alone clinic at each facility so that patients who did not want integrated care would be able to continue with usual clinics”.

We do not think it is within our remit to discuss re-purposing HIV clinics or building new primary care clinics. We conducted a feasibility study on integrated management of chronic conditions. The question of whether HIV clinics should be re-purposed or new Centres should open can not be asked at this early stage of the research. This is also a question that we believe policy-makers and not researchers will be asking.

Authors should look to shorten their introduction, the second paragraph (Lines 16-25) seems unnecessary. Also, what is the main question of the paper? Is it that, will people seek care at an integrated health clinic that also serves HIV, despite possible stigma? The lack of a clear question causes confusion throughout the rest of the paper

RESPONSE: We have clarified the question that this research addressed, as mentioned above. Lines 16-25 gives an important historical context of why HIV care is provided from separate stand-alone conditions. We agree to cut out the 2 lines below as some of this is touched upon in the discussion:

‘Separate stand-alone clinics for each chronic condition are unlikely to be affordable for governments and hugely problematic for patients with multiple conditions’

Methods

The methods are unfortunately excessive. Authors should consider moving most of methods to supplementary material or re-consider what is necessary to include in the manuscript in order for results to be replicated

Thank you for this observation: We have revised and shortened the methods section into two pages

Page 5, Lines 23-Line 31 seems unnecessary

Page 5 Lines 46-Page 6 line 58 – should be moved to supplementary or condensed

Page 7 Lines 21-41 – this seems unnecessary to have

Retention needs an explicit definition. Is it considered retention if a patient has one follow up visit at 6 months?

RESPONSE: This is the first study of its kind. The methods is long as we are dealing with both health systems issues (e.g. types of health facilities studied) and patient issues across different diseases (e.g. how blood pressure, fasting glucose and plasma viral load were measured).

However, in light of the referees comments, we have cut out the section on stakeholder engagement and integrated key aspects of this into the patient and public involvement section. This is on page 8 and reads:

The concept of integrated care was developed in partnership with senior policy makers. In each country, we held planning meetings with the Ministry of Health and non-governmental partners at national and local level, with senior clinicians and hospital managers and with patient leaders and community representatives. National steering committees were formed, comprising representatives from these stakeholders together with the researchers and these committees contributed to the research strategy and guided its implementation. We also formed an international steering committee, that included 4 senior researchers and 4 independent researchers, which had oversight of the whole programme. We also held investigator meetings involving all of the partners and these included patient representatives and policy-makers.

Retention does have a explicit definition. It is the first sentence of the statistical analysis.

If blood pressure and glycemc control are important measures of success to this project, there should be mention of this in the introduction

RESPONSE: We state this in the introduction as you suggested. This reads:

We conducted a cohort study of the integrated care for HIV-infection, diabetes and hypertension and evaluated its feasibility and acceptability in terms of retention in care and clinical indicators including control of blood pressure and fasting plasma glucose.

Authors should not feel the need to mention specific instruments used to measure A1C, record blood pressures, or measures Page 8 lines 3-22 can largely be removed or condensed

RESPONSE: In our view it is standard practice to state the instruments used for the biomedical tests. There are very few data on the background levels of blood pressure and fasting plasma glucose in such populations and readers will need to know which instruments were used to be able to interpret the test results.

Results

Authors should re-evaluate what results should be included in the manuscript. Would advise to only

include results that are pertinent to their original question. For example, providing relative risk ratios for obesity among those with diabetes vs HIV, seems irrelevant.

Response: One key objective is to examine clinical indicators by disease status and that's why we describe the distribution of obesity by disease status.

Figure 1 – too large, should be condensed. Would reason there is no need to stratify by disease

Response: The diseases for which we have integrated care have historically very different rates of retention. For example less than 10% of people with diabetes are estimated to be in regular care compared with 90% of people with HIV. It is essential that we stratify by disease status.

Table 2 and Table 4 need to be combined to one table for easy interpretation.

Response: Table 2 shows baseline data and describes the cohort. Table 4 shows the outcome data at the study end (i.e. shows the effect of integration). We have not compared changes with baseline directly. In our view, it would not be logical to combine these two tables as they serve different purposes.

Discussion

Study makes comparisons to other studies that assessed retention. Would be important to state if retention is defined differently in other studies

Authors should look to condense discussion.

RESPONSE. We have used a common definition of retention (which the other studies used also). We also show all the data to enable readers to calculate retention in different ways.

The difference in retention rates between our study and those reported elsewhere are very large but we have been careful not to over-interpret this. There are many other potential confounders and differences between these studies that makes comparisons difficult, and we do not wish to spend more time discussing the different ways in which retention could be defined.

This a complex study covering multiple diseases and issues at both patient and health system level. We believe that condensing the discussion further would lose valuable information.

A point to be made for that is in reference to the entire manuscript - it is thought to be stigmatizing to describe patients with HIV, as living with "HIV infection." Would recommend changing verbiage to reflect the more accepting terminology, "People/Persons/Patients with HIV".

Response: People living with HIV is commonly used (and in fact has a acronym – PLWHIV which is commonly used).

"In our study, individuals with more than one condition, who would normally attend multiple different clinics,"

1. This is a statement that should be in the introduction as it gets across the importance of authors' work

Response: The referee is asking us to cut the introduction and to introduce new themes into it. It is crucial that integration is for all people with (target) chronic conditions and not just for those with multiple conditions. Indeed for those with multiple conditions, integrated care in Africa is clearly attractive. The real question is whether people with single conditions should be managed in an integrated clinic, which is where we have focussed our research (and this paper).

VERSION 2 – REVIEW

REVIEWER	Morgan Birabaharan University of California San Diego
REVIEW RETURNED	02-Oct-2021

GENERAL COMMENTS	The authors submit a revised version of their manuscript “Integrating health services for HIV-infection, diabetes, and hypertension in sub-Saharan Africa: a cohort study” Lines 94: Consider specifying what “done in close to normal health service conditions” may mean. Lines 177-181: Can consider moving to appendix or removing from manuscript Lines 315: Can consider specifying what loss to follow up entailed (i.e. no visit within 6 months). Would it be correct to interpret any person that was enrolled and had a visit within 6 months was considered “retained in care?” Lines 339: Can consider specify what conditions “very sick” may entail. Also consider specifying what “other reasons” may be Lines 361-362: “HIV-infection versus” comparison group is missing in this statement Lines 443: Consider adding a citation for this hypothesis
---

VERSION 2 – AUTHOR RESPONSE

Lines 94: Consider specifying what “done in close to normal health service conditions” may mean.

RESPONSE. This is in the abstract. We have changed this to ‘done in close to real-life health service conditions.

Lines 177-181: Can consider moving to appendix or removing from manuscript

RESPONSE: We have removed this as the reviewer suggested

Lines 315: Can consider specifying what loss to follow up entailed (i.e. no visit within 6 months). Would it be correct to interpret any person that was enrolled and had a visit within 6 months was considered “retained in care?”

RESPONSE: We have clarified this. This now reads: A patient was regarded as lost to follow up if they had not attended an appointment within the last 6 months.

Lines 339: Can consider specify what conditions “very sick” may entail. Also consider specifying what “other reasons” may be

Response: We have clarified that by sick we mean requiring urgent medical attention or hospital admission. The ‘other’ conditions for ineligible were just 5. For ‘other’ who declined, who were n=18, we say that this was ‘**mostly that they needed to be in specialist care**’

Lines 361-362: “HIV-infection versus” comparison group is missing in this statement

Response: we have clarified this. This now reads:

Obesity was more common among participants with diabetes, hypertension or multimorbidity than among participants with HIV (504/1441 (35.0%) versus 116/832 (13.9%),

Lines 443: Consider adding a citation for this hypothesis.

RESPONSE. This reads ‘The mixing of patients with HIV-infection with those living with diabetes and hypertension may also have led to the sharing of experiences of living with a long-term condition, which could have brought further benefit’.

This is not based on a citation. Please note that this is the first ever study conducted of integrated care of this nature.